# The Impact of Economic Growth Target Constraints on Environmental Pollution: Evidence from China

**DOI:** 10.3390/ijerph20042831

**Published:** 2023-02-06

**Authors:** Changfei Nie, Wen Luo, Yuan Feng, Zhi Chen

**Affiliations:** 1School of Economics and Management, Nanchang University, Nanchang 330031, China; 2College of City Construction, Jiangxi Normal University, Nanchang 330022, China; 3School of Economics and Trade, Hubei University of Economics, Wuhan 430205, China

**Keywords:** economic growth target, goal constraint, environmental pollution, sustainable development, China

## Abstract

Economic growth target (EGT) has become an essential tool for macroeconomic administration all around the world. This study examines the effect and mechanisms of EGT on environmental pollution (EP) by using economic growth target data from provincial Government Work Reports in China from 2003–2019. The conclusions denote that EGT significantly aggravates regional EP, and it still stands after robustness tests and instrumental variable (IV) estimation. The result of mediating effect shows that EGT aggravates EP mainly from three ways: investment surge, technological innovation, and resource allocation. The result of the moderating effect shows that government’s fiscal space positively adjusts the effect of EGT on EP, while environmental regulation negatively adjusts the effect of EGT on EP. The heterogeneity test reflects that the effect of EGT on EP is more significant on provinces that adopt a “hard constraint” setting method and fulfill EGT. Our study provides a reference to better balance the link between EGT and sustainable development for the government department.

## 1. Introduction

The growing EP situation has caused many negative impacts on food security, social economy development, and land health condition, being a global challenge that threatens human survival and sustainable development in all countries [1,2]. The global characteristics of the EP problem have attracted worldwide attention, more than 150 countries and regions have announced a carbon neutrality target and carbon peaking target. The Chinese government is no exception. General Secretary Xi Jinping presented an ambitious aim “striving for carbon peaking by 2030 and carbon neutrality by 2060” at the 75th UNGA (United Nations General Assembly) in September 2020. The aim not only reflects China’s determination to promote global environmental governance and maintain sustainable development, but also demonstrates the determination to construct a community of common destiny with all countries in the world. However, China’s long-standing extensive economic growth model has caused a serious EP problem and the ecological environment has been increasingly deteriorating. The 2022 Global Environmental Performance Index reported that the EPI score of China is 28.4, ranked 160th out of 180 countries, far below Denmark (77.9), the UK (77.7), and Finland (76.5). The 2022 BP World Energy Statistics Yearbook reflects the carbon emissions of global regions, among which, China’s total carbon emissions are 10,523 megatons and the carbon emissions growth rate is as high as 5.8%, ranking at the top in both total carbon emissions and growth rate. The development trend will continue, which is detrimental to the green development of the world. China’s EP problems not only impede national economy development but also cause irreparable damage to citizens’ physical and psychological health, severely limiting the achievement of sustainable development and the aim of high-level economic development [3,4,5].

The deterioration of the EP situation forces all countries to accelerate discussions and research on topics such as sustainable development and circular economy (CE) development, and the development of CE is gradually becoming the common view in many countries [6]. From 2014–2016, the number of circular economy articles published increased from 30 per year to 100 per year [7]. The Regulation on Waste issued by the European Community in 2008 stressed that waste management is an important aspect of CE, and 4R (reduce, reuse, recycle, recover) are identified as the key of CE policy implementation [8,9]. Additionally, many great consulting firms such as Deloitte, Ernst & Young, and McKinsey also published CE reports and white papers [10,11,12].

From the perspective of world trade, UNCTAD (United Nations Conference on Trade and Development) presented that in today’s highly economic globalization, the management of the global resource cycle entails the cooperation of all countries to maintain the circularity of products in international trade. As the world’s second-largest economy, China has engaged in the upstream and downstream of global value chains and caused a significant impact on the global resource cycle. Moreover, the Chinese increasingly promoted high-quality economic development and also focused on the future of CE. As a result, the highest managers continue to elevate CE’s strategic status, eventually recognizing it as a national strategic policy. China’s CE policy began during the Hu-Wen administration. In 2005, the state council of China issued Several Opinions on the Development of Circular Economy, which clarified the guiding ideology, fundamental principles, and primary goals of CE. The Circular Economy Promotion Law of the People’s Republic of China issued in 2008 was a milestone in the development of China’s CE at the legal level. In 2015, the CPC Central Committee and the State Council issued Opinions on Accelerating the Construction of Ecological Civilization, stressing that green development, circular development, and low-carbon development are the main ways to construct China’s ecological civilization, denoting that the CE of China has stretched into all aspects of society. In the meantime, the development conditions of Chinese society and the national development strategy provided “ground” for the implementation of CE. The Eleventh Five-Year Plan (2006–2010) and Twelfth Five-Year Plan (2011–2015) take building a resource-saving and environment-friendly society as one of the important tasks of development. The 14th Five-Year Plan (2021–2025) entails constructing a modern economic system, promoting high-quality economic development, and promoting green development. Although the strategic thinking behind the development of CE has been fully carried out, it is hard to measure the development level of CE in different regions. Referring to Pesce et al., this paper creates a conceptual model of CE development and judges the distribution and development level of CE in different regions of China based on six indicators: system thinking, innovation, stewardship, collaboration, value optimization, and transparency [13].

Since 1950, China has consistently announced EGT and used EGT to macro-manage national economy development. From the view of the EGT setting method, Chinese governments formulated EGT of China’s Five-Year Plan through NPC (The National People’s Congress). The Chinese government also declares the year’s national EGT in its Government Work Report and gives guidance to each provincial government on economic development, decomposing the national EGT to local governments in different levels through the administrative hierarchy system. However, a majority of local governments set a high EGT compared with higher-level governments. It can be explained that China’s vertical governance model makes “layer by layer phenomenon” and “top-down competition for scales” commonplace [14,15]. It has an irrepressible amplification effect on EGT, namely, that EGT set by local governments are higher than those set by central government. Hence, under China’s top-down management system and vertical governance model, to obtain the qualifications for promotion, government officials must pay more attention to the achievement of EGT, thus, the target can be easily executed in China. 

Figure 1 illustrates the provincial average EGT as generally higher than the national EGT from 2003 to 2019, demonstrating that EGT is amplified through “layer by layer phenomenon.”

Figure 2 further reports the gap in provincial EGT and national EGT from 2003 to 2019, which shows that the majority of the provincial EGT are higher than national targets, and the gap is mainly concentrated in the range of “0–2%” and “2–4%”, further reflecting the phenomenon of “layer by layer.”

The constraints of EGT will further influence local government behavior. For one thing, EGT setting level directly represents the local government’s confidence in economic construction of jurisdiction. EGT achieved or not will largely affect the promotion and political future of government officials [16], which can strengthen the work enthusiasm of government officials, improve their work efficiency, and motivate officials to fulfill EGT perfectly. For another aspect, EGT has an important constraining effect on local government behavior. To achieve EGT, local governments probably adopt some extreme measures to obtain rapid economic growth. For instance, implementing “racing to bottom” strategies in environmental regulation even sacrifice the environment to achieve EGT [17]. As a result, under EGT constraints, local governments may allocate resources to projects that are conducive to boosting economic growth, and the behavior may further influence EP, which is the primary concern of this paper. 

Specifically, we propose the theoretical hypotheses from existing relevant economic theories, and empirically test the research hypotheses by manually collecting EGT data announced in the Government Work Report from 2003–2019 for 30 provinces in China. The baseline estimation results indicate EGT significantly aggravates EP, and the finding is robust. These basic conclusions are not substantially changed after IV estimation and many robustness tests, such as changing measurement methods of main variables, eliminating the non-linear relationship, eliminating the sample of municipalities, adding additional covariates, and eliminating the effect of outliers. The mediating effect results denote that EGT significantly increases local investment level, inhibits innovation development, lower resource allocation efficiency, and thus exacerbates EP. The moderating effect results show that the higher government’s fiscal space level will make the effect of EGT on EP greater. The stronger environmental regulation intensity will make the effect of EGT on EP smaller. In the end, the heterogeneity tests show the impact of EGT on EP is more significant in provinces that employ “hard constraint” in EGT formulation and achieve EGT.

Compared to existing research, the main contributions of the paper are as follows: First, in terms of the research topic. Even if several scholars have studied the effect of EGT [18,19], few studies specifically examine the link between EGT and EP. The research conducted by Chai et al. and Wang et al. is closest to this paper but mainly focuses on air pollution problems and carbon emission situations, respectively [20,21]. In this paper, we comprehensively build an environmental pollution index (EPI) from three dimensions of pollution structure to reflect the EP conditions in each province. Moreover, we further investigate the channels of EGT aggravating EP and consider the moderating effect of EGT on EP. It is an effective addition to existing research. Second, in terms of theoretical contribution, the traditional goal-setting theory mainly focuses on the incentive effect of goals on micro-individuals [22,23], but this paper identifies the impact of goal-setting from the macro aspect—EGT set of government, which expands the theoretical boundary of goal-setting theory to a certain extent. Third, in terms of practical significance, the macro-control of national economic growth through the setting of EGT is not unique to China, and it has been a global phenomenon. From the 1950s, the number of economies which announced EGT gradually increased and peaked in the 1970s. The number of economies which have announced EGT showing a renewed upward trend in the beginning of the 21st century. According to statistics, up till now, at least 110 economies around the world have announced or are still announcing growth targets, such as European countries, India, China, Vietnam, and Japan [24]. Hence, the conclusions of the paper can provide a reference for government departments of each country to effectively balance economic growth and EP, and to scientifically set EGT. 

The rest of this paper is as follows. Section 2 is a literature review. Section 3 presents a theoretical analysis and research hypotheses. Section 4 describes the model, variables, and data. Section 5 shows the empirical results and analysis. Section 6 is a complementary analysis, including the test of mediating effect, moderating effect analysis, and heterogeneity analysis. Section 7 summarizes research conclusions and presents relevant policy recommendations.

## 2. Literature Review

### 2.1. The Impact of EGT

Being a core factor guiding economic development, EGT is indispensable in national macroeconomic development. In recent years, the impact of EGT has started to obtain more and more attention from scholars.

Wang examined the direct effect of EGT on economic growth, and found that as a special government behavior, the formulated EGT is an essential external driving to boost the economic growth [25]. Some scholars focus on the effect of EGT on other macroeconomy aspects. For instance, Liu used EGT data from 230 Chinese cities from 2003 to 2016, and showed that the high EGT may reduce public service expenditures, which will inhibit long-term economic growth [26]. Borge denoted EGT will cause a severe resource misallocation phenomenon, namely, that an increase in fiscal expenditure can significantly improve rapid economic growth while reducing financial expenditures that enhance long-term economic growth [27]. Lin and Zhang found EGT constraint will increase pressure on government officials to achieve the formulated EGT, and hence, significantly influence local governments’ policy on land transfer [28]. Furthermore, some scholars examine the impact of EGT on enterprises. For example, Li presented that EGT set by local governments can significantly inhibit the growth of enterprises’ technological innovation level [22]. Zhong showed that the high-pressure EGT will force local officials to deregulate the supervision of environment, which enable enterprises to reduce environmental protection investment [29]. Zhao and Cheng demonstrated that the pressure of economic growth will decrease the enterprises’ TFP by distorting the marginal productivity of capital [23].

With a sustainable development notion gradually taking root in the public heart, a few scholars have begun to study the link between EGT and green development. For example, Shen demonstrated that the “top-down amplification” and the “layer by layer” effect have an inhibition impact on the improvement of green technology innovation levels [30]. Li showed that the enhancement of EGT will significantly diminish environmental regulation intensity, which is detrimental to the economy and society’s sustainable development [31]. Fan examined the relationship between EGT and carbon emissions, showing that EGT is positively correlated with carbon emissions [32]. Sun demonstrated that EGT has a significant inhibition impact on green total factor (GTFP), and this inhibition effect is more prominent in provinces which have fulfilled the target [33].

### 2.2. The Influencing Factors of EP

Factors affecting EP are a classical topic in environmental economics research. In general, scholars studied environmental regulation [34,35], economic development [36,37], energy consumption [38], economic agglomeration [39], and foreign direct investment [40,41,42]. Among them, the most relevant research topic to this paper is the link between economic growth and EP. A lot of studies have proved the existence of environmental Kuznets curve (EKC). For example, Shahbaz successfully demonstrated the existence of EKC by using Turkey’s national economic data from 1970 to 2010 [43]. Fodha and Zaghdoud demonstrated that there is an inverse U-shaped link between pollutant emissions and economy growth, showing the existence of EKC [44]. However, other scholars have found EKC does not exist. For instance, Adu and Denkyirah showed that the link between EP and economy growth is not significant, namely, that the EKC does not exist [45]. Özokcu and Özdemir conducted an empirical study, and the model results are N-shaped and inverse N-shaped relationship, and the results do not support the existence of EKC [46].

To summarize the above studies, we can find that although there exists abundant literature that concerns the connection of economic growth and EP, there are few studies that combine both EGT and EP. Although a few of the latest literature has begun to focus on relevant topics, the research is mainly examined from the perspectives of environmental target setting [47,48], green technology innovation [30], and air pollution [21], and does not research the effect of EGT on EP deeply, which provides space for the research of this paper.

## 3. Theoretical Analysis and Research Hypothesis

### 3.1. The Impact of EGT Constraints on EP

EGT management is an instrument of macroeconomic control by governments around the world. EGT setting has a positive incentive impact on government officials at all levels, which drives the officials to meet or even overachieve the pre-set EGT, and subsequently sends a strong “ability signal” to the high-level governments and eventually increases the possibility of political promotion [49,50]. However, to reach the formulated EGT, government officials will compete with colleagues, and local governments will seek rapid economic growth at the expense of environment [17]. Moreover, in a fiscal decentralization system, the regional government has greater power to allocate financial resources. To fulfill the formulated EGT, regional governments may distort financial resources allocation [51], which will inhibit the enhancement of technological innovation levels, and it is detrimental to EP regulation. Furthermore, the constraint of EGT will increase regional governments’ financial pressure, and raise the tendency of regional governments to “seek development with the land”, which will lead to an increase in the area of land granted. However, the majority of the land granted is used to construct heavy industrial enterprises, which brings serious EP problem to society [52,53]. In addition, although local governments can attract foreign capital through large-scale investment attraction to realize rapid economic development, from a sustainable development aspect, foreign direct investment will build a large number of pollution-intensive enterprises. In this way, it will form a “pollution heaven” in the jurisdiction area, increase wastewater and toxic waste emissions, and contribute to a severe EP problem [54,55,56]. After summing up the analysis above, we present the hypothesis:

**H1:** *EGT can aggravate the EP*.

### 3.2. The Channels of EGT Aggravates EP

First, the investment surge effect. In essence, the EGT management system is an assessment of GDP growth rate during the tenure of local officials. Since fixed asset investment is the key impetus of economy development and has an increasingly strong driving power for GDP growth, most government officials are choosing to increase fixed asset investment to significantly improve “GDP performance” and to achieve the formulated EGT [57,58,59]. During the limited tenure time, local government officials continuously expand the investment in the jurisdictions, send “investment signals” to outside capital markets, and build a favorable investment environment and conditions to motivate the emergence of investment convergence, eventually realizing the “surge effect” of fixed asset investment [60]. However, the rapid growth of fixed asset investment accelerates the pollution-intensive enterprises’ development, increases fossil fuels’ consumption, reduces fossil fuels’ utilization efficiency, and brings a large amount of wastewater and waste gas, which will generate many ecological environment problems and seriously damage the sustainable development of society [61,62,63]. Therefore, EGT may increase the level of fixed asset investment and further aggravate EP.

Second, the technological innovation effect. EGT management is mainly concerned with economic growth during one’s tenure. To fulfill the pre-set EGT, government officials will allocate more resources to projects or fields, which may enhance economic growth during a short time, when allocating fiscal resources, and thus generate the “squeeze out effect” on education and technology fields, and inhibit the enhancement of the technological innovation level in the jurisdictional area [53]. The environmental situation will be further affected when the technological innovation level in the jurisdiction is suppressed. However, some studies have demonstrated technology innovation will lower haze pollution, increase energy utilization efficiency, and reduce air pollution and industrial waste pollution [64,65,66]. Therefore, EGT may inhibit the improvement of technological innovation levels and further aggravate EP.

Third, the resource allocation effect. The paper explains resource allocation effect on EP in two ways: the condition of resource allocation and the efficiency of resource utilization. For one view, under the fiscal decentralization system, government officials have strong control over the local finances [67,68]. To overachieve EGT and increase the political promotion possibility, government officials are more likely to allocate financial resources to productive departments of the economy, interfere with the normal resource allocation of factor markets, break the spontaneous resource flow of the market economy, and protect several heavy polluted enterprises from being eliminated, seriously distorting the resource allocation situation [69,70,71]. For another view, under the distorted resource allocation, the market interest rate increases due to the destruction of the supply–demand balance in the financing market, which contributes to the increase of enterprises’ financing cost and loan cost, and make resources that cannot be exchanged effectively between enterprises and market. Facing economic growth pressure, regional governments may rise fiscal expenditure on fields such as infrastructure and fixed asset investments, which will hinder the transformation and structural upgrading of related industries, resulting in resources allocation imbalance, decreasing the resource utilization efficiency of industries [72,73]. Therefore, EGT may distort resource allocation and further aggravate EP. After summing up the analysis above, we present the hypothesis:

**H2:** *EGT can aggravate EP through investment surge effect, technological innovation effect, and resource allocation effect*.

### 3.3. The Moderating Mechanism of EGT Aggravates EP

(1)Government’s fiscal space. In regional economic development, government departments participate in most of the activities related to economic growth and have greater intervention in regional GDP growth [74,75]. For reaching the formulated EGT as soon as possible, the government will use the relevant policy system to adjust economy development. For one way, when regional economic growth lacks motivation, government departments tend to increase investment expenditure in the secondary industry and allocate more resources to protect the development of regional resource-dependent industries, and attract foreign direct investments to spur economy’s rapid growth [26]. For another way, when local governments have excessive pressure on economic growth, they will squeeze out fiscal expenditures in fields such as education and science and technology that promote long-term economic development and increase the area of land transfer to achieve EGT during one’s tenure [76]. No matter if it is rising the secondary industry’s investment or enlarging the land transfer area, a series of government intervention will increase the emission of regional pollutant gases and pollutants, which is against the concept of sustainable development and aggravates regional EP. Therefore, a higher government’s fiscal space level may rise the effect of the EGT aggravating EP.(2)Environmental regulation. Many researches demonstrate the combination of environmental regulation and regional economy development may effectively descend EP level. On one hand, government can effectively promote the improvement of regional technological innovation speed and improve the green technology innovation capacity by using environmental regulation [77]. In the meanwhile, environmental regulation can constrain local governments’ financial resource allocation, maximize the efficiency of financial resource utilization, and drive local government officials to focus on social sustainable development, eventually reducing the damage of economic development on environmental protection [78,79]. On the other hand, an appropriate environmental regulation intensity can motivate the enterprise innovation behavior and enable enterprises to realize structure optimization and adjustment, which will push industrial structure transformation and upgrading [80,81]. Ai demonstrated that environmental regulation was beneficial to improve the industrial production situation, reduce the total greenhouse gas emissions, and increase the energy utilization efficiency [82]. Environmental regulation may stimulate the enterprises’ “innovation compensation”, enhance enterprises’ competitiveness capacity, and drive industrial structures’ adjustment, optimization, and upgrading [83,84]. Hence, a higher environmental regulation intensity may decrease the effect of the EGT aggravating EP. After summing up the analysis above, we present the hypothesis:

**H3:** *Government’s fiscal space will positively adjust the effect of EGT on EP, and environmental regulation will negatively adjust the effect of EGT on EP*.

Figure 3 depicts the EGT aggravating EP channels and moderating mechanisms.

## 4. Model and Data Description

### 4.1. Model Specification

In order to explore the impact of EGT on EP, this paper set the basic regression mode:(1)$$EPI_{it}=\beta_{0}+\beta_{1}Target_{it}+\varphi X_{it}+\mu_{i}+\eta_{t}+\varepsilon_{it}$$
where the subscript *t* and *i* express the year and province, respectively; *EPI* is the explained variable, reflecting EP level; *Target* is the core independent variable, referring to EGT; *X* presents all control variables; *μ_i_* and *η_t_* represent province fixed effect and year fixed effects, respectively, and *ε_it_* is error term. In this paper, the coefficient of *β*_1_ is essential, and according to research Hypothesis 1, we expect *β*_1_ to be significantly positive.

### 4.2. Variables

#### 4.2.1. Explained Variable

Environmental pollution index (EPI) is the explained variable in this paper. Considering that it is difficult for a single pollutant emission index to reflect the EPI comprehensively, accurately, and objectively [85], therefore, we establish an index system to measure the EPI, and embody the multidimensional characteristics of EP. Specifically, we chose three indicators: per capita industrial sulfur dioxide (SO_2_) emissions, per capita industrial sewage emissions, and per capita industrial smoke and dust emissions to measure EPI by using the entropy method. The process of calculation is as follows:

First, to eliminate the effects of the order of magnitude and the dimension of the original data, we standardize the original data:(2)$$z_{ij}=\frac{x_{ij}-\min(x_{ij})}{\max(x_{ij})-\min(x_{ij})}$$
where *x_ij_* denotes the per capita pollutant emissions of category *j* in province *i*; max(*x_ij_*) and min(*x_ij_*) represent the maximum and minimum values of different types of per capita pollutant emissions in all provinces, respectively; *z_ij_* denotes the standardized value.

Second, calculate the entropy value according to the following formula:(3)$$E_{j}=-1/\ln n\sum_{i=1}^{n}p_{ij}\ln p_{ij},\;where\;p_{ij}=z_{ij}/\sum_{i=1}^{n}z_{ij}$$

Third, calculate each indicator weight:(4)$$w_{j}=(1-E_{j})/\sum_{j=1}^{m}\left(1-E_{j}\right)$$

Finally, calculate *EPI* according to the following formula:(5)$$EPI_{i}=10\times \sum_{j=1}^{m}w_{j}\times z_{ij}$$
where *EPI* represents the EP index of each region, and its value ranges from 0 to 10. The EP is more serious when the value is higher.

#### 4.2.2. Core Independent Variable

The core independent variable in this paper is economic growth target (EGT), and it is measured by using the data of EGT published in the provincial Government Work Report over the years. For the few EGT published in the form of interval, we use mean values to express (for example, Hainan Province formulated an EGT of 7–7.5% in 2019, and we use the mean value of 7.25%). Moreover, among research samples in this paper, Shanxi Province in 2006, Henan Province in 2011, and Shanghai city in 2015 did not publish a clear EGT, hence we use the goal proposed in the five-year plan as a substitute in this paper.

#### 4.2.3. Control Variables

For further eliminating the possible effects of omitted factors on estimation results, we refer to relevant literatures to control the following variables: (1) Economic growth (*Pgdp*), which is measured by logarithm of per capita real GDP (constant price in 2003) [86]; (2) Industrial structure (*Indus*), represented by the ratio of the added value of secondary and tertiary industries to GDP [87]; (3) Urbanization level (*Urban*), calculated by the ratio of urban population in each province to total population [32]; (4) Human Capital Level (*Hc*), denoted by the average years of educational years [88]; (5) Infrastructure construction (*Infra*), calculated by per capita road area [85]; (6) Openness level (*Open*), represented by the ratio of total value of imports and exports to GDP [89,90], and the total value of imports and exports is converted to RMB by using the average exchange rate during the year; (7) Population density (*Popden*), calculated by logarithm of population per square kilometer [91].

### 4.3. Data Sources and Descriptions

In this paper, we chose China’s 30 provinces from 2003–2019 as the sample (some indicators of Tibet were removed because of the missing data), and the relevant data were mainly collected from the China Statistical Yearbook, China Regional Economic Statistical Yearbook, and EPS Database. The sudden outbreak of the COVID-19 pandemic had an impact on China’s economic development and environmental pollution, and the relationship between EGT and EP will be interfered. To avoid the overlap with the COVID-19 pandemic, we ending the sample period in 2019. The descriptions and definitions of variables are shown in Table 1. It is shown that the EPI in each province from the sample data range from 0.05 to 7.80, indicating that in different provinces, there exists a great spatial difference of EP level. Meanwhile, EGT published in each provincial Government Work Report range from 4.5% to 15%, with an average value of 9.438%, which shows that the local governments have a greater economic growth pressure.

## 5. Empirical Results and Analysis

### 5.1. Fitted Curve Analysis

In order to observe the relationship between EGT and EP directly, we draw a fitted curve between them in this paper. The observation in Figure 4 illustrates there exists a significant positive correlation between EGT and EP, which indicates EGT is formulated higher, EPI is bigger and EP is more serious. However, the research conclusion based on graphical observation cannot fully prove the cause and effect between EGT and EP. Therefore, we will further identify this finding in the following with a rigorous econometric model.

### 5.2. Basic regression

The paper utilizes stepwise regression to examine the basic regression model, and Table 2 shows the regression results. In column (1), the estimation coefficient of *Target* is positive and significant without adding any control variables, which indicates EGT can significantly aggravate the EP levels, namely, EGT will increase EPI in each province. After gradually adding control variables from *Pgdp* to *Popden*, the coefficient size and significance of *Target* do not change significantly, which further reflect that EGT aggravates regional EP conditions. The estimation results in column (8) show that the coefficient of *Target* is 0.103, and it is significant on 1% level, which demonstrates that EGT can aggravate regional EP again. Therefore, EGT will cause damage to the environment, and it is detrimental to the enhancement of environmental development and society’s sustainable development.

From the regression results of control variables, it is showing that the estimation coefficient of *Hc* is negative and significant, which indicates that the increase of *Hc* will reduce EPI, and is similar with the conclusion of Jun [92] and Lan [93]. The coefficient of *Urban* is also significantly negative, and it may be explained by the fact that the urbanization development in China accelerates innovative cities’ construction and promotes cities’ green development, which will reduce the EP levels [94,95]. Moreover, the estimation coefficient of *Pgdp*, *Indus*, *Infra*, *Open*, and *Popden* are not significant.

### 5.3. Robustness Tests

In order to ensure the reliability of baseline regression results, this paper conducts robustness tests: (1)Changing explained variable measurement method. The paper uses the equal weight method to assign weights to the indicators, namely that the weights of per capita industrial SO_2_ emissions, per capita industrial wastewater emissions, and per capita industrial smoke and dust emissions are set to 1/3; employs equal weight method to re-measure each province’s EPI, and finally re-estimates the regression results [91]. The regression results are expressed in Table 3, column (1). The coefficient of *Target* is 0.014 and significant in 1% level, which indicates that EGT positively affects EPI and therefore aggravates the regional EP.(2)Changing core independent variable measurement method. EGT reflects the economy growth pressure faced by local governments. We combine the characteristics of “layer by layer” of China’s EGT [26], construct economy growth pressure index as a core independent variable, and use *Pressure* variable to re-estimate estimation results. The calculation formula of economy growth pressure index is:
(6)$$Pressure=\left(Target-NTarget\right)/NTarget$$
where *Target* and *NTarget* represent EGT published by provinces and country, respectively; *Pressure* reflects the “overweight” degree of each province compared with national EGT. The estimation results are expressed in Table 3, column (2). The coefficient of *Pressure* is 0.780 and is significant, indicating that the economy growth pressure index aggravates EP and increases EPI level. 

(3)Eliminating the non-linear relationship. Considering EGT and EP may have a non-linear relationship [21], we further add the squared term of EGT (*Target_sq*) based on the basic regression and re-estimate estimation results. The estimation results are expressed in Table 3, column (3). The coefficient of *Target_sq* is −0.010, but it is not significant, which indicates EGT and EP do not have a non-linear relationship, and the finding supports the baseline regression results of this paper.(4)Eliminating samples from municipalities. Considering that municipalities are quite different from ordinary provinces in terms of administrative hierarchy and other fields, this paper further eliminates four municipalities: Tianjin, Beijing, Chongqing, and Shanghai, and re-estimates regression results. The estimation results are expressed in Table 3, column (4). The coefficient of *Target* is 0.099 and is significant in 1% level, which indicates that after eliminating samples from municipalities, EGT still aggravates EP and will increase the level of EPI.(5)Adding additional covariates. Angrist and Pischke demonstrated that by adding interaction term of time-trending variables with control variables in a regression model can effectively control the time trend of influencing factors of the explained variable [96], and it can also alleviate the bias of regression estimation caused by other factors over time to a certain extent. Thus, we further add the interaction term of time-trending variables with control variables to the basic model and re-estimate the regression results. The estimation results are expressed in Table 3, column (5). The coefficient of *Target* is 0.064 and is significant in 5% level, which indicates that after adding additional covariates, EGT will also increase the level of EPI.(6)Eliminating the effects of outliers. To remove the effect of a small number of possible outliers in the sample on estimation results, this paper treats all ordered variables with upper and lower 1% Winsorized in this paper and re-estimates the regression results. The estimation results are expressed in Table 3, column (6). The coefficient of *Target* is 0.110 and is significant in 1% level, indicating that after eliminating the effects of outliers, EGT will also cause damage to the environment and increase the EPI level.

**Table 3 ijerph-20-02831-t003:** Robustness test.

Variables	(1)	(2)	(3)	(4)	(5)	(6)
*Target*	0.014 ***		0.303 **	0.099 ***	0.064 **	0.110 ***
	(0.004)		(0.147)	(0.027)	(0.025)	(0.024)
*Pressure*		0.780 ***				
		(0.176)				
*Target_sq*			−0.010			
			(0.007)			
Control variables	√	√	√	√	√	√
Year FE	√	√	√	√	√	√
Province FE	√	√	√	√	√	√
*Obs*	510	510	510	442	510	510
*R* ^2^	0.562	0.588	0.589	0.575	0.641	0.591

Note: Standard errors in parentheses, *, **, and *** represent significance levels of 10%, 5%, and 1%, respectively.

### 5.4. Instrumental Variable (IV) Estimation

The above estimation results may face an endogenous threat to EGT. Specifically, the endogeneity problem is mainly derived from two ways. One way is the omitted variables. Although the study controls some potential variables that will affect EPI, there will still be other variables that have an impact on EPI, which means that this paper cannot fully resolve the problem generated by omitted variables. Another way is the two-way cause and effect. There exists a two-way cause and effect between EGT and EPI, namely that the EGT has an impact on EPI while EPI also has an impact on EGT. Therefore, for further elimination of the endogeneity problem caused by omitted variables and two-way causal relationships, this paper uses the IV estimation method to estimate the regression results.

Referring to Shen [30], this paper uses an interaction item between the amount of prefecture in different provinces and national-level EGT with a one-period lag as an IV and re-estimates the regression results. The logic behind our setting the IV is that the amount of prefecture in each province largely reflects the regional government competition level, and the number of jurisdictions in a province is more, whereas the local government competition level is greater. In order to moderate the competition level in different local governments and form an effective incentive for lower-level governments, the higher-level governments tend to decompose a relatively lower EGT to lower-level governments. In other words, the number of jurisdictions in the province is negatively correlated with EGT [15], and therefore meets the choice of instrument which satisfies the “inclusion restriction”. Meanwhile, since the number of prefectures in the province is a fixed value, it does not influence the regional EP level directly, and thus satisfies the requirement of the exogeneity of IV.

The results of IV estimation are shown in Table 4. Under the first phase of regression, the coefficient of *IV* is negative and significant in 1% level, and corresponds to the expected estimation results. At the same time. The Cragg–Donald Wald F statistic are bigger than 10, indicating that the weak IV problem does not exist. In the second phase of regression, the coefficient of *Target* is positive and significant in 5%, whether or not control variables are added, and the result denotes that the main conclusion of this paper remains unchanged after considering the endogeneity problem between EGT and EP.

## 6. Further Analysis

### 6.1. Mechanisms Tests

The above research results in this paper suggest that EGT significantly aggravates EP. To deeply investigate the inner mediating effect of EGT aggravating EP, this paper employs a mediating effect model to examine, and builds specific models as below:(7)$$EPI_{it}=\beta_{0}+\beta_{1}Target_{it}+\varphi X_{it}+\mu_{i}+\eta_{t}+\varepsilon_{it}$$
(8)$$Med_{it}=\gamma_{0}+\gamma_{1}Target_{it}+\varphi X_{it}+\mu_{i}+\eta_{t}+\varepsilon_{it}$$
(9)$$EPI_{it}=\theta_{0}+\theta_{1}Target_{it}+\theta_{2}Med_{it}+\varphi X_{it}+\mu_{i}+\eta_{t}+\varepsilon_{it}$$
where *Med* is the mediating variable, and the notion of other variables corresponds to the basic model. The test steps of the mediating effect model are as follows: First, estimate model (7) and obtain total effect *β*_1_ of EGT on EP. Second, estimate Equations (7) and (8) and observe the significance of coefficient of *γ*_1_ and coefficient of *θ*_2_. Third, observe the size of the coefficient of *θ*_1_ in Equation (9). If *θ*_1_ is significant and has the same sign as *β*_1_ and the absolute value of *θ*_1_ is smaller than the absolute value of *β*_1_, it indicates the mediating effect is existing. Relying on the estimation results of baseline regression, *β*_1_ is 0.103 and significant in 1% level. Therefore, in the subsequent mechanism effect test, we mainly estimate Equations (8) and (9), observe the significance of coefficient of *γ*_1_ and coefficient of *θ*_2_, and judge the sign of *θ*_1_ and *β*_1_ as well as the relative size of the absolute value of *θ*_1_ and *β*_1_.

This paper tests the theoretical H2 from three ways: investment surge effect, technological innovation effect, and resource allocation effect. Specifically, this paper uses the ratio of fixed asset investment to GDP (*Invest*) to represent investment scale [62]; uses the amount of invention patent granted per 10,000 people (*Inno*) to denote technological innovation [97]; uses the total factor productivity (*Tfp*) to express resource allocation efficiency [98], and *Tfp* is calculated by Solow residual method.

The regression results of the mediating effects model are expressed in Table 5. The estimation results in column (1) illustrate that the coefficient of *Target* is positive and significant in 1% level, indicating the higher EGT, the higher the *Invest*, namely, that EGT will increase the scale of fixed asset investment. The results in column (2) demonstrate that the increase of fixed asset investment can significantly increase the level of EPI. Meanwhile, the coefficient of *Target* in column (2) is 0.054, and the absolute value is lower than 0.103, expressing that EGT will increase the level of EPI through promoting the investment scale. Therefore, the investment surge effect is proved. 

The estimation results in column (3) indicate the coefficient of *Target* is −0.363 and significant, which indicates the formulation of EGT will inhibit the improvement of regional technology innovation level and is not conducive to the development of technology innovation capability. The regression results in column (4) denote that the coefficient of *Inno* is negative and significantly in 1% level, which indicates technological innovation will alleviate regional EP and improve regional environmental conditions. Meanwhile, the coefficient of *Target* in column (4) is 0.096, and the absolute value is lower than 0.103, which indicates that EGT will aggravate EP and increase the level of EPI by inhibiting the enhancement of regional technology innovation level, proving that the technological innovation effect is existing.

The regression results in column (5) suggest the coefficient of *Target* is negative and significant, which demonstrates that the existence of EGT will distort resource allocation and damage the normal resource flow in the regional factor market. The results in column (6) indicate that the coefficient of *Tfp* is −0.962 and significant in 1% level, which indicates that the efficient allocation of regional resources can reduce the level of EPI and is conducive to social sustainable development. Meanwhile, the coefficient of *Target* is 0.090, and the absolute value is lower than 0.103, which indicates that EGT can increase EPI level by reducing resource allocation efficiency and distorting resource allocation, proving the existence of the resource allocation effect.

### 6.2. Moderating Mechanism Analysis

The effect of EGT on EP will probably be influenced by other factors. Therefore, we further examine the effect from two aspects, government’s fiscal space and environmental regulation, and construct the model as follows:(10)$$EPI_{it}=\beta_{0}+\beta_{1}Target_{it}+\beta_{2}Mor_{it}+\beta_{3}Target_{it}\times Mor_{it}+\varphi X_{it}+\mu_{i}+\eta_{t}+\varepsilon_{it}$$
where *Mor* is the moderating variable, and the notion of other variables corresponds to the basic model. Government’s fiscal space (*Gov*) is represented by the ratio of general government fiscal expenditures to GDP [99]. On the one hand, the government’s fiscal conditions will influence government spending multipliers and credit provision directly and indirectly [100]. Meanwhile, increasing government spending may limit the government’s ability to intervene. On the other hand, a large amount of evidence indicates that higher government fiscal expenditures will cause increased economic activity [101,102]. On the contrary, higher levels of government fiscal spending imply less government regulation behavior [103]. Therefore, an increase in government fiscal expenditures will reduce the capacity of government intervention regulation, thereby amplifying the overall effect. Environmental regulation (*Er*) is denoted by the ratio of completed investment to industrial pollution control [17]. Over the past decades, the ESG (Environmental, Social, Governance) idea has drawn more and more attention from the social and academic sectors. Dantas denoted that the relevance of the ESG factor for institutional investors increased [104]. Additionally, ESG factors are particularly relevant for the Chinese economy because the developed economies’ central banks introduced a battery of quantitative easing policies to increase economic mobility during 2008–2017 [105,106]. Chinese governments are no exception. They also integrate the ESG idea into daily regulation to keep the environment condition in good condition while motivating economic development. Hence, the implementation of environmental regulation can not only ensure the sustainable development of the social economy but also restrain the behavior of environmental pollution, and finally attenuate the overall effect.

Table 6 reports the results of the moderating mechanism. The results in column (1) suggest the coefficient of *Target* × *Gov* is 0.028 and is significant, demonstrating that the greater the government regulation, the more damage of EGT on the environment. The reason is, the special fiscal decentralization model of China gives local officials more rights to handle the fiscal expenditures, and under the “GDP-centered” promotion assessment system, local officials probably employ the method of “racing to the bottom” to reduce the protection of the regional ecological environment to seek political promotion [107,108]. 

The results in column (2) denote that the coefficient of *Target* × *Er* is 0.633 and is significant in 1% level, illustrating the higher the environmental regulation level, the lower the damage of EGT on the ecological environment. It can be explained in two ways: in one way, the implementation of environmental regulation by government departments can push forward urban industrial transformation and development, which is beneficial to industrial structure upgrading, effectively reducing the pollution emission of the industry and having a positive effect on social sustainable development and social environmental protection [109]. In another way, environmental regulation will increase enterprises’ GTFP level, reducing the emissions of polluting gases, and decreasing the damage of industry production on environment [110]. 

### 6.3. Heterogeneity Analysis

The economy growth acts as the key factor in the performance evaluation model. Since central government directly links the economy growth with the political promotion of officials, and the higher-level government will compulsorily formulate some economic growth indicators, and therefore differences exist in the constraint level of EGT generated by different local government. Meanwhile, there are differences in infrastructure investment, industrial structure, and land transfer in different regions, and therefore there are differences in the completed situation of EGT by government officials in different regions. Therefore, this paper conducts a heterogeneity analysis of the research sample according to the different EGT constraint levels and the different EGT completion situations. The estimation results are expressed in Table 7.

(1)Heterogeneity in EGT constraint levels. The government departments have the typical characteristics of “hard constraint” and “non-hard constraint” when formulating EGT. The “hard constraint” refers to when setting EGT, using adverb expressions, such as “ensure”, “above”, and “strive”; while the “non-hard constraint” refers to using adverb expressions such as “up and down”, “around”, and “between” when formulating EGT [111]. All levels of government in China are placed in a top-down target incentive model, namely, the formulation of the goal of economy growth in lower-level governments is always influenced by the formulation of the goal of economy growth in higher-level governments through “layer by layer” effect, and there also exists a “top-down competition for scales” in the formulation of EGT [112,113]. Under the “promotion tournament” system, governments at all levels will adopt a “hard constraint” situation to formulate EGT and use the adverb expressions “ensure” and “above” to emphasize the annual target of economic growth.

Relying on “hard constraint” and “non-hard constraint”, we classify the sample data into two groups. The regression results are expressed in Table 7, columns (1) and (2). The results indicate that for provinces which employ “hard constraint” in EGT setting, the coefficient of *Target* is 0.289 and significant in 1% level; while for provinces which employ “non-hard constraint” in EGT setting, the coefficient of *Target* is 0.087, and also significant in 1% level. The results demonstrate that EGT set by “hard constraint” is indeed more harmful to the ecological environment than EGT formulated by “non-hard constraint”. The results perhaps can be explained as follows: EGT set by “hard constraint” will increase the pressure on government departments to fulfill the economy growth goal and force the government to mainly focus on promoting rapid economy growth, which leads to insufficient investment in environmental protection. While EGT set by “non-hard constraint” will effectively reduce the pressure on government departments to achieve the growth target, enabling governments to develop effective solutions to promote social economy growth and maintain social sustainable development under the situation of “allow leeway.”

(2)Heterogeneity in the completed situation of EGT. Theoretically, whether EGT is fulfilled or not plays a decisive role in officials’ promotion. If local officials want to meet the political promotion requirements, they must obtain rapid economic growth to achieve the formulated EGT and earn the “promotion ticket” [76]. In the meantime, the financial expenditure in ecological environment development and environmental regulation will be relatively reduced, and this will aggravate EP. On the contrary, if local officials want to meet the synergistic development of environmental protection and economic growth, then government departments will allocate financial resources to national education, enterprise technological innovation and industry structure upgrading to seek a long-term economic development [114], and therefore reduce the damage of EGT on the environment. The solution will help governments achieve social and economic sustainable development. However, under such a way, local government officials may have difficulty in meeting the formulated EGT and even lose the “promotion ticket.”

This paper regards “the gap of real GDP growth rate minus EGT” as the indicator to measure the completed situation of EGT. If the indicator value is lower than 0, it is regarded as “not complete EGT, otherwise it is regarded as “complete EGT.” Based on the achievement or not, the research divides the sample data into two types, and the estimation results are expressed in Table 7, columns (3) and (4). The results denote that, for provinces that fulfill EGT, the coefficient of *Target* is 0.091 and significant in 1% level. While for provinces that do not fulfill EGT, although the coefficient of *Target* is positive, it is not significant, which indicates that the damage of EGT on the environment generally exists in provinces that fulfill EGT. The reasons can be expressed as below: the provinces that achieve EGT tend to distribute more resources into economy growth and reduce resources in the enterprise’s green technology innovation and environmental regulations, which inhibits the increase of the regional green development level. While the provinces that do not fulfill EGT tend to focus on the shared development between social sustainable development and economic sustainable development, and rationally allocate the financial resources in economy development and environmental protection, which reduce the damage of EGT on environment.

## 7. Conclusions and Policy Implications

The Chinese government has consistently used EGT to manage economic growth in the process of economic development for a long time. EGT acts as an essential position in increasing governance enthusiasm of local government officials and promoting the rapid growth of the local economy. And the influence of EGT gradually stretches from economic performance to other areas of economic development, further affecting the sustainable development and ecological environment protection of China. Under this background, this paper manually collects EGT data from provincial Government Work Reports in China from 2003–2019, and examines the impact and mechanisms of EGT on EP. The conclusions are as follows: (1) The basic regression results demonstrate that EGT can significantly aggravate EP and increase the provincial EPI level, and this finding still stands after IV estimation and robustness tests such as changing the measurement method of explained variable and core independent variable, eliminating the non-linear relationships, eliminating samples from municipalities, adding additional covariates, and eliminating the effects of outliers, demonstrating the robustness of paper findings. (2) The mediating effect results show EGT will significantly increase the regional fix asset investment scale, inhibit technology innovation, and decrease resource allocation efficiency. (3) The test of the moderating mechanism finds that government’s fiscal space and environmental regulation have an impact on the effect of EGT on EP. The government’s fiscal space is higher, the effect of EGT on EP is greater. The degree of environmental regulation is stronger, the effect of EGT on EP is smaller. (4) There is a heterogeneity in the impact of EGT on EP for provinces that adopt “hard constraint” to formulate EGT, and provinces that achieve the formulated target will significantly aggravate EP. To summarize the above conclusions, we suggest some policy recommendations as follows:

First, weaken the relationship between EGT and the promotion assessment system and rationally formulate EGT for different regions. The central government should not solely focus on high GDP growth rates when formulating EGT. Instead, it should combine the requirements of social sustainable development, and balance the link between economy development and environmental development. Furthermore, the central government should also strengthen ecological civilization construction to prevent excessive EGT from aggravating regional EP. Meanwhile, the local governments ought to rationally formulate EGT and avoid “layer by layer” based on EGT announced by the higher-level governments. Therefore, Chinese governments should lower the weight of GDP indicators in government officials’ performance assessment systems and establish a comprehensive performance assessment system that includes environmental indicators such as sustainable development and environmental protection level. In addition, the central government should also make appropriate adjustments according to the actual development and economic level of different regions, apply different assessment systems to officials in different provinces depending on local conditions, and continuously improve the government officials’ assessment system to increase the fairness and reasonableness of promotion.

Second, improve the impact of the investment surge effect, technological innovation effect, and resource allocation effect on EP. From the perspective of the investment surge effect, government departments should control the investment scale, optimize the investment structure, improve investment efficiency, formulate investment agreements that are beneficial to environmental development, and avoid the blind investment attraction behavior that aims at achieving the set EGT. From the view of technological innovation effect, government departments should rationally allocate the limited resources, promote the financial resources’ allocation efficiency, and ensure financial resources will be allocated to the fields of technological innovation, science and technology, education, and other fields that promote long-term economic growth. Moreover, government departments should also encourage the effective innovation of enterprises, formulate relevant policies to promote the innovation enthusiasm of all sectors of society, and create a positive innovation atmosphere to better achieve the sustainable development of society. From the perspective of resource allocation effect, government departments should maintain the resource allocation status of the factor market and ensure the spontaneous resource flow in the market. In addition, to improve the flow rate of resources between enterprises and the market, government departments should also increase the utilization efficiency of social resources, and accelerate the transformation and structural upgrading of relevant industries.

Third, decrease government intervention level and increase environmental regulation intensity. For one way, government departments should reduce or avoid intervening in the normal economy development to prevent the distorted allocation of fiscal expenditures, and decrease the positive adjustment of government’s fiscal space on the EGT pollution effect. Meanwhile, we should fully utilize the market’s proactive effect in the process of social resource allocation and make resources spontaneously circulate among various departments of society. For another way, government departments should increase environmental regulation intensity in economic development to push jurisdictional enterprises to accelerate the speed of green technology innovation, eliminate pollution-intensive industries, and realize the economy and society’s sustainable development.

Fourth, set the regional EGT according to different regional development conditions, instead of setting EGT with a “one-size-fits-all” method. The findings of this paper indicate that the “hard constraint” on EGT formulation drives government officials to focus on economic growth while neglecting environmental protection and development and aggravates regional EP under the influence of “layer by layer.” In contrast, the “non-hard constraint” on EGT formulation will reduce the pressure of economic growth on government, improve the total factor productivity level under the setting method of “allow leeway”, and decrease the impact of EGT on EP. At the same time, the provinces that achieve EGT have a significantly destructive impact on the environment, while provinces that do not fulfill the target have a non-significant destructive effect on the environment. Therefore, the central government cannot uniformly set the target while formulating EGT for different regions. It should carefully consider the current state of different regional developments, analyze the favorable directions for future development of different regions, integrate the local existing resources to set EGT based on local conditions, and finally realize the win-win situation of economic growth and sustainable social development.

## Figures and Tables

**Figure 1 ijerph-20-02831-f001:**
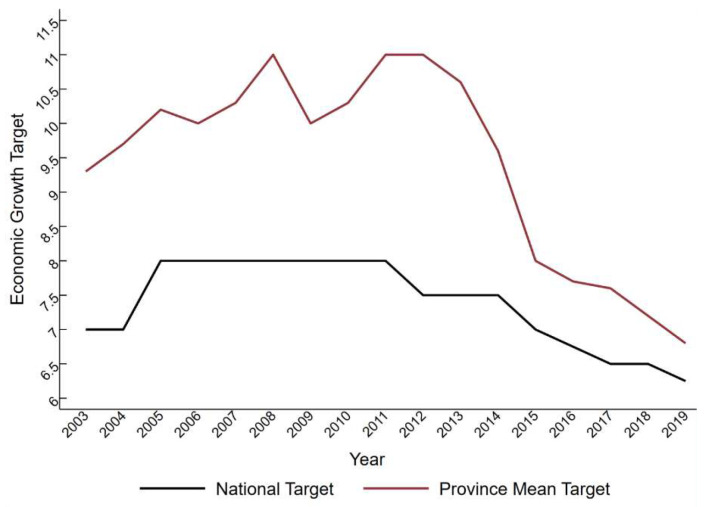
National EGT and provincial average EGT.

**Figure 2 ijerph-20-02831-f002:**
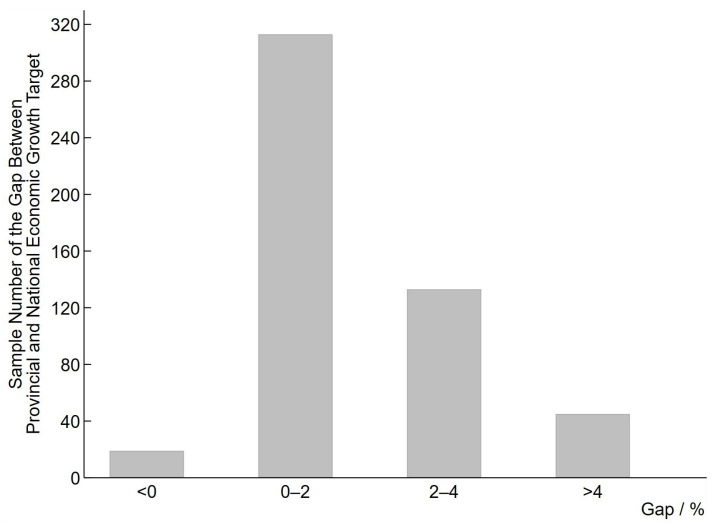
The gap between provincial EGT and national EGT.

**Figure 3 ijerph-20-02831-f003:**
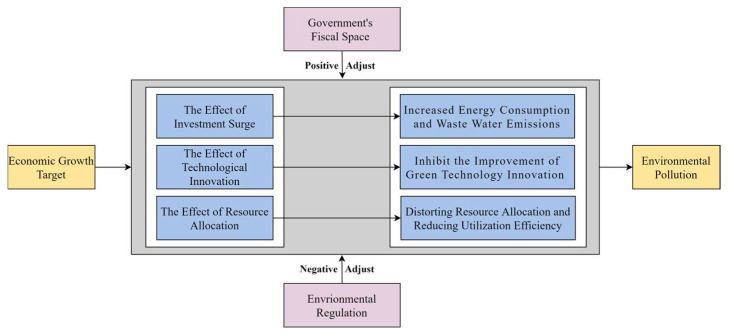
The channels and moderating mechanism of EGT aggravating EP.

**Figure 4 ijerph-20-02831-f004:**
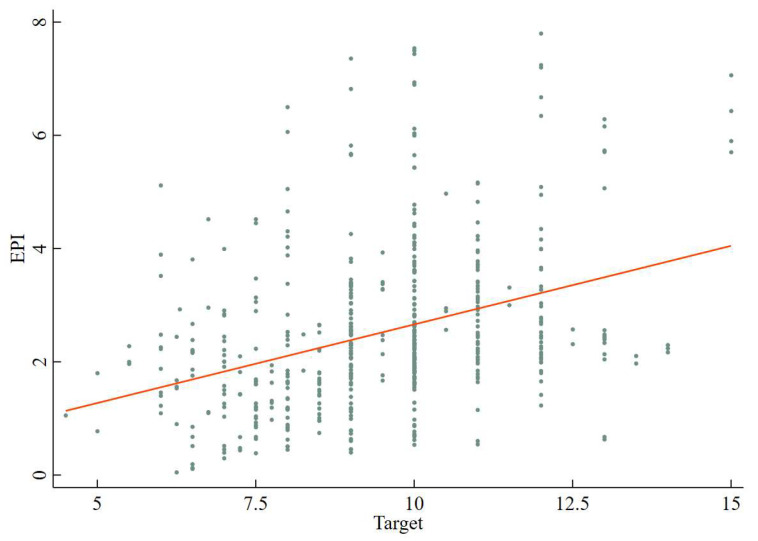
The fitted curve between EGT and EP.

**Table 1 ijerph-20-02831-t001:** Definitions and descriptions of variables.

Variables	Definitions	Obs	Mean	SD	Min	Max
*EPI*	Environmental pollution index	510	2.504	1.439	0.050	7.800
*Target*	Economic growth target disclosed in Government Work Report	510	9.438	1.840	4.500	15.000
*Pgdp*	Logarithm of per capita real GDP (constant 2003 prices)	510	10.107	0.752	8.190	12.122
*Indus*	The ratio of the added value of secondary and tertiary industries to GDP	510	0.887	0.061	0.630	0.997
*Urban*	The ratio of urban population to the total population	510	0.528	0.146	0.139	0.896
*Hc*	Average years of schooling	510	8.721	1.024	6.040	12.782
*Infra*	Per capita road area	510	13.435	4.680	4.040	26.196
*Open*	The ratio of the total value of imports and exports to GDP	510	0.309	0.376	0.013	1.722
*Popden*	Logarithm of population per square kilometer	510	7.696	0.626	5.226	8.749

**Table 2 ijerph-20-02831-t002:** Basic regression results.

Variables	(1)	(2)	(3)	(4)	(5)	(6)	(7)	(8)
*Target*	0.088 ***	0.097 ***	0.100 ***	0.096 ***	0.103 ***	0.104 ***	0.104 ***	0.103 ***
	(0.023)	(0.023)	(0.024)	(0.023)	(0.024)	(0.024)	(0.024)	(0.024)
*Pgdp*		−0.732 *	−0.613	−0.387	−0.229	−0.326	−0.322	−0.329
		(0.415)	(0.449)	(0.445)	(0.448)	(0.473)	(0.474)	(0.475)
*Indus*			−1.068	−1.140	−1.444	−1.490	−1.297	−1.224
			(1.555)	(1.529)	(1.526)	(1.529)	(1.571)	(1.592)
*Urban*				−2.112 ***	−2.145 ***	−2.092 ***	−2.048 ***	−2.037 ***
				(0.520)	(0.518)	(0.525)	(0.531)	(0.533)
*Hc*					−0.314 **	−0.313 **	−0.321 **	−0.318 **
					(0.130)	(0.130)	(0.131)	(0.132)
*Infra*						0.008	0.010	0.009
						(0.012)	(0.013)	(0.013)
*Open*							−0.125	−0.128
							(0.231)	(0.232)
*Popden*								0.019
								(0.065)
*Constant*	2.026 ***	8.683 **	8.472 **	7.310 *	8.589 **	9.414 **	9.278 **	9.127 **
	(0.232)	(3.779)	(3.793)	(3.742)	(3.760)	(3.980)	(3.991)	(4.028)
Year FE	√	√	√	√	√	√	√	√
Province FE	√	√	√	√	√	√	√	√
*Obs*	510	510	510	510	510	510	510	510
*R* ^2^	0.563	0.566	0.567	0.582	0.587	0.587	0.588	0.588

Note: Standard errors in parentheses, *, **, and *** represent significance levels of 10%, 5%, and 1%, respectively.

**Table 4 ijerph-20-02831-t004:** IV estimation.

Variables	(1)	(2)	(3)	(4)
	*Target*	*Target*	*EPI*	*EPI*
*IV*	−0.066 ***	−0.063 ***		
	(0.019)	(0.018)		
*Target*			0.330 **	0.330 **
			(0.151)	(0.154)
Control variables		√		√
Year FE	√	√	√	√
Province FE	√	√	√	√
*Obs*	510	510	510	442
*R* ^2^	0.562	0.588	0.589	0.575

Note: Standard errors in parentheses, *, **, and *** represent significance levels of 10%, 5%, and 1%, respectively.

**Table 5 ijerph-20-02831-t005:** Mediating effect tests.

Variables	(1)	(2)	(3)	(4)	(5)	(6)
	*Invest*	*EPI*	*Inno*	*EPI*	*Tfp*	*EPI*
*Target*	0.037 ***	0.054 **	−0.363 **	0.096 ***	−0.014 **	0.090 ***
	(0.005)	(0.024)	(0.169)	(0.024)	(0.006)	(0.023)
*Invest*		1.30 ***				
		(0.210)				
*Inno*				−0.020 ***		
				(0.007)		
*Tfp*						−0.962 ***
						(0.187)
Control variables	√	√	√	√	√	√
Year FE	√	√	√	√	√	√
Province FE	√	√	√	√	√	√
*Obs*	510	510	510	510	510	510
*R* ^2^	0.744	0.620	0.752	0.596	0.887	0.610

Note: Standard errors in parentheses, *, **, and *** represent significance levels of 10%, 5%, and 1%, respectively.

**Table 6 ijerph-20-02831-t006:** Moderating effect tests.

Variables	(1)	(2)
*Target*	0.029	−0.007
	(0.041)	(0.032)
*Gov*	3.269 **	
	(1.366)	
*Target* × *Gov*	0.228 *	
	(0.138)	
*Er*		−5.898 ***
		(1.319)
*Target* × *Er*		0.633 ***
		(0.129)
Control variables	√	√
Year FE	√	√
Province FE	√	√
*Obs*	510	510
*R* ^2^	0.622	0.611

Note: Standard errors in parentheses, *, **, and *** represent significance levels of 10%, 5%, and 1%, respectively.

**Table 7 ijerph-20-02831-t007:** Heterogeneity analysis.

Variables	(1)	(2)	(3)	(4)
	Hard Constraint	Non-Hard Constraint	Fulfill Target	Not Fulfill Target
*Target*	0.289 ***	0.087 ***	0.091 ***	0.093
	(0.074)	(0.026)	(0.033)	(0.057)
Control variables	√	√	√	√
Year FE	√	√	√	√
Province FE	√	√	√	√
*Obs*	112	398	351	159
*R* ^2^	0.664	0.616	0.546	0.692

Note: Standard errors in parentheses, *, **, and *** represent significance levels of 10%, 5%, and 1%, respectively.

## Data Availability

Not applicable.

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
