# Peer review of "The Impact of Economic Growth Target Constraints on Environmental Pollution: Evidence from China"

_ijerph, 2023, doi:10.3390/ijerph20042831_

Round 1
Reviewer 1 Report
1. Introduction is very weak and it should be enhanced by adding some 2022 and 2023 references. The main idea, novelties, and also, research gap are not considered.
2. Add a topic in the field of circular economy in the paper. In my opinion, it is very important when you want to talk about eco-environmental evaluations.
3. Consider below paper and create a conceptual model for CE distribution in different places of China.
https://www.mdpi.com/2071-1050/12/3/832
4. At least evaluate the policies of CE in China. It is the main aspects of your research.
https://www.sciencedirect.com/science/article/abs/pii/S0195925520301943
5. You should consider UNCTAD as one of your main ideas and explain how can the targets can be executed in China.
https://unctad.org/topic/trade-and-environment/circular-economy
Author Response
Response to Reviewer 1 Comments
Manuscript ID: ijerph-2193561
Title: The impact of economic growth target constraints on environmental pollution: Evidence from China
Journal: International Journal of Environmental Research and Public Health
Dear Editor and Reviewer,
Thank you for your comments and constructive suggestions that help us to improve the manuscript. We are now submitting a revised version of the manuscript for review. All the revisions have been marked in red font for your convenience. Please find our point-by-point responses to all the comments below.
Point 1: Introduction is very weak and it should be enhanced by adding some 2022 and 2023 references. The main idea, novelties, and also, research gap are not considered.
Response 1: Thanks for your suggestion. Follow your advices, on the one hand, we have added some references published in 2022 and 2023 to enhance the quality of Introduction (See lines 27-29;45-48) .On the other hand, we have compared with other articles which the topic is similar with this study to fully express the novelties, main idea, and research gap from three ways (See lines 139-157).
Point 2: Add a topic in the field of circular economy in the paper. In my opinion, it is very important when you want to talk about eco-environmental evaluations.
Response 2: Thanks for your suggestion. After carefully thinking the link between EP (environmental pollution) and CE (circular economy), we added a topic of CE field in this paper (See lines 49-85) .
Point 3: Consider below paper and create a conceptual model for CE distribution in different places of China. https://www.mdpi.com/2071-1050/12/3/832
Response 3: Thanks for your suggestion. Refer to above paper, we created a China different regions’ conceptual model of CE distribution based on the principles of CE (See lines 80-85) .
Point 4: At least evaluate the policies of CE in China. It is the main aspects of your research.
https://www.sciencedirect.com/science/article/abs/pii/S0195925520301943.
Response 4: Thanks for your suggestion. In the paragraphs of CE topic, we carefully evaluated and listed the policies of CE, and introduced the behavior of CE on national strategic level (See lines 61-80) .
Point 5: You should consider UNCTAD as one of your main ideas and explain how can the targets can be executed in China. https://unctad.org/topic/trade-and-environment/circular-economy
Response 5: Thanks for your suggestion. We have added UNCTAD as one of the ideas in this paper based on the UNCTAD dedicated to keep the balance between trade and environment. (See lines 58-61) And we explained how the targets can be executed in China through the governance model and officials’ promotion(See lines 93-99).

Reviewer 2 Report
Please refer to my attached comments.

Author Response
Response to Reviewer 2 Comments
Manuscript ID: ijerph-2193561
Title: The impact of economic growth target constraints on environmental pollution: Evidence from China
Journal: International Journal of Environmental Research and Public Health
Dear Editor and Reviewer,
Thank you for your comments and constructive suggestions that help us to improve the manuscript. We are now submitting a revised version of the manuscript for review. All the revisions have been marked in red font for your convenience. Please find our point-by-point responses to all the comments below.
Summary. This paper examines the effect of economic-growth targets (EGT) on environmental pollution (EP). The authors use economic growth target data from provincial Government Work Reports from China during the sample period 2003–2019. Overall, the authors find that EGT aggravates regional EP, a conclusion that is robust to a battery of robustness tests. The authors also estimate the moderating effects of government fiscal capability and environmental regulation to shed light on the economic significance of the channels proposed in the manuscript.
Overall assessment. My overall assessment is that the research question is worthy of consideration for publication given its relevance for academics and policymakers. Generally speaking, the paper is well executed and the authors do not over-claim the extent of their findings.
Despite the positive outlook, I believe the present version of the paper is silent about important channels through which the moderating variables considered amplify or attenuate the overall effect.
As such, for the paper to be publishable in an academic outlet such as the International Journal of Environmental Research and Public Health, the authors must (at least) accomplish the following. I urge the authors to consider these points carefully, including in their revision the references to the relevant literature.
Point 1: The role of the government’s fiscal conditions. When introducing the “government intervention” measure (gov), I encourage the authors to refer to it as “government’s fiscal space” or simply “government spending.” Observe that government spending as a fraction of GDP is typically one of the dimensions of the fiscal space of a jurisdiction (see Kose et al., 2022). Still in the paragraph where the authors introduce the moderating variables, the authors should provide the following justifications with the appropriate references.
–The fiscal conditions of the government, directly and indirectly, affect the provision of credit and government spending multipliers (see Silva, 2021). As such, elevated spending may imply that the government’s ability to intervene is potentially compromised.
–In fact, there is vast evidence that higher government spending is associated with increased economic activity (see Nakamura and Steinsson, 2014; Chodorow-Reich, 2019). Conversely, high levels of government spending mean that governments’ regulatory activity may be compromised (see Cowx et al., 2022).
Response 1: Thanks for your suggestion. We have done such revision: first, use “government’s fiscal space” to replace “government intervention” when introducing the “government intervention” measure in this paper(See lines 17;133;291;305;325;569;700;701;744). Second, after referring to the relevant literature provided, we explained the channels through which government’s fiscal space amplifies the overall effect (See lines 573-581) .
Point 2: The importance of ESG factors. When introducing the variable “environmental regulation” (Er), the authors should discuss that over the past decade, the relevance of environmental, social, and governance (ESG) factors for institutional investors increased massively (refer to Dantas, 2021).
–This is particularly relevant for the Chinese economy because the 2008–2017 period is characterized by massive injections of liquidity by central banks of developed economies through a series of quantitative easing policies (refer to Dedola et al., 2020; Cortes et al., 2022). This should be also mentioned after introducing the ‘environmental regulation” variable.
Response 2: Thanks for your suggestion. We have found the setback and added the relevant references provided. We have clarified the effect of ESG factors on Chinese economic development, and explained the channels through which environmental regulation attenuates the overall effect (See line 581-593).
Point 3: Additional comments. The authors should provide a clearer justification for their sample ending in 2019. A plausible reason could be simply that it is important to avoid any overlap with the COVID–19 pandemic because it ultimately affected economic growth and environmental pollution.
Response 3: Thanks for your suggestion. We have added an expression to explain why the sample period ending in 2019 (See lines 388-391) .
Point 4: Instead of calling it “requirement of IV correlation,” the authors should refer to this statistical association using the technical term—the choice of instrument satisfies the “inclusion restriction.”
Response 4: Thanks for your suggestion. We have changed the “requirement of IV correlation” into technical term—the choice of instrument satisfies the “inclusion restriction” (See lines 501) .
References
Chodorow-Reich, G. (2019). Geographic Cross-Sectional Fiscal Spending Multipliers: What Have We Learned? American Economic Journal: Economic Policy 11(2), 1–34. DOI: https://doi.org/10.1257/pol.20160465.
Cortes, G. S., G. P. Gao, F. B. Silva, and Z. Song (2022). Unconventional Monetary Policy and Disaster Risk: Evidence from the Subprime and COVID–19 Crises. Journal of International Money and Finance 122, 102543.DOI: https://doi.org/10.1016/j.jimonfin.2021.102543.
Cowx, M., F. B. G. Silva, and K. Yeung (2022). Government Deficits and Corporate Tax Avoidance. Available at SSRN 4060416. DOI: https://dx.doi.org/10.2139/ssrn.4060416.
Dantas, M. (2021). Are ESG Funds More Transparent? Available at SSRN 3269939. DOI: http://dx.doi.org/10.2139/ssrn.3269939.
Dedola, L., G. Georgiadis, J. Gräb, and A. Mehl (2020). Does a Big Bazooka Matter? Quantitative Easing Policies and Exchange Rates. Journal of Monetary Economics. https://doi.org/10.1016/j.jmoneco.2020.03.002.
Kose, M. A., S. Kurlat, F. Ohnsorge, and N. Sugawara (2022). A Cross-Country Database of Fiscal Space. Journal of International Money and Finance 128, 102682. DOI: https://doi.org/10.1016/j.jimonfin.2022.102682.
Nakamura, E. and J. Steinsson (2014). Fiscal Stimulus in a Monetary Union: Evidence from US Regions. American Economic Review 104(3), 753–92. DOI: https://doi.org/10.1257/aer.104.3.753.
Silva, F. B. G. (2021). Fiscal Deficits, Bank Credit Risk, and Loan-Loss Provisions. Journal of Financial and Quantitative Analysis 56(5), 1537–1589. DOI: https://doi.org/10.1017/S0022109020000472.

Round 2
Reviewer 1 Report
In my opinion, this paper can be accepted.